# Arrhythmogenic Right Ventricular Cardiomyopathy in Children: A Systematic Review

**DOI:** 10.3390/diagnostics14020175

**Published:** 2024-01-12

**Authors:** Stefana Maria Moisa, Elena Lia Spoiala, Eliza Cinteza, Radu Vatasescu, Lacramioara Ionela Butnariu, Crischentian Brinza, Alexandru Burlacu

**Affiliations:** 1Pediatrics Department, Faculty of Medicine, “Grigore T. Popa” University of Medicine and Pharmacy, 700115 Iasi, Romania; stefana-maria.moisa@umfiasi.ro; 2“Sfanta Maria” Clinical Emergency Hospital for Children, 700309 Iasi, Romania; ionela.butnariu@umfiasi.ro; 3Pediatrics Department, Faculty of Medicine, “Carol Davila” University of Medicine and Pharmacy, 700115 Bucharest, Romania; elizacinteza@yahoo.com; 4“Marie Curie” Clinical Emergency Hospital for Children, 41451 Bucharest, Romania; 5Cardio-Thoracic Department, “Carol Davila” University of Medicine and Pharmacy, 020021 Bucharest, Romania; 6Clinical Emergency Hospital, 050098 Bucharest, Romania; 7Genetics Department, Faculty of Medicine, “Grigore T. Popa” University of Medicine and Pharmacy, 700115 Iasi, Romania; 8Faculty of Medicine, “Grigore T. Popa” University of Medicine and Pharmacy, 700115 Iasi, Romania; alexandru.burlacu@umfiasi.ro; 9Institute of Cardiovascular Diseases “Prof. Dr. George I.M. Georgescu”, 700503 Iasi, Romania

**Keywords:** arrhythmogenic right ventricular dysplasia, cardiomyopathy, children

## Abstract

Arrhythmogenic right ventricular cardiomyopathy (ARVC) is an inherited disease characterized by the progressive replacement of the normal myocardium by fibroadipocytic tissue. The importance of an early diagnosis is supported by a higher risk of sudden cardiac death in the pediatric population. We reviewed the literature on diagnosis, risk stratification, and prognosis in the pediatric population with ARVC. In case reports which analyzed children with ARVC, the most common sign was ventricular tachycardia, frequently presenting as dizziness, syncope, or even cardiac arrest. Currently, there is no gold standard for diagnosing ARVC in children. Nevertheless, genetic analysis may provide a proper diagnosis tool for asymptomatic cases. Although risk stratification is recommended in patients with ARVC, a validated prediction model for risk stratification in children is still lacking; thus, it is a matter of further research. In consequence, even though ARVC is a relatively rare condition in children, it negatively impacts the survival and clinical outcomes of the patients. Therefore, appropriate and validated diagnostic and risk stratification tools are crucial for the early detection of children with ARVC, ensuring a prompt therapeutic intervention.

## 1. Introduction

Arrhythmogenic right ventricular dysplasia/cardiomyopathy (ARVD/C), also known as arrhythmogenic cardiomyopathy, is an inherited disease characterized by the progressive replacement of the normal myocardium by fibro-adipocytic tissue, mainly in the right ventricle, but in advanced phases also in the left ventricle and the atria [1]. Although rare, affecting only 0.05% of the general population [2], ARVD/C is one of the biggest challenges for cardiologists due to the severe complications which can occur—ventricular arrhythmia, heart failure, and even sudden death [3].

While the first scientific observation of a possible ARVD/C case series may be dated back to the early 18th century, when the Italian physician Giovanni Maria Lancisi published a four-generation analysis of heart disease recurrence with palpitations, heart failure, right ventricle aneurysms, and sudden cardiac death in multiple family members [4], it was only in 2000 that McKoy et al. identified the involvement of mutations in gene encoding for plakoglobin. This desmosomal protein plays an essential role in the resistance of cardiac cells to mechanical stress [5].

Currently, numerous genetic determinants and phenotypic manifestations have been discovered in ARVD/C. Most cases are inherited as an autosomal dominant trait, with incomplete penetrance and variable expressivity, but isolated cases, apparently sporadic, have also been cited [6]. The importance of an early diagnosis is supported by a higher risk of sudden cardiac death in the pediatric population, probably due to more frequent myocarditis-like episodes with consequent electrical instability [7].

Understanding the main features of ARVD/C in the pediatric population is important due to several reasons. The early detection and diagnosis of ARVC in pediatric patients can lead to timely interventions and appropriate management strategies. Children with ARVD/C are at an increased risk of life-threatening arrhythmias and sudden cardiac death, making accurate diagnosis and effective treatment crucial for their well-being and long-term outcomes. Secondly, studying ARVD/C in children allows for a better understanding of the unique clinical features and manifestations of the disease in this specific population. By focusing on pediatric cases, researchers and healthcare professionals can gain insights into the specific challenges and nuances associated with ARVD/C in children, leading to improved diagnostic accuracy.

Furthermore, investigating ARVD/C may contribute to a broader understanding of the genetic basis and pathophysiology of the disease. ARVD/C is known to be associated with a pathogenic or likely pathogenic gene variant in more than 60% of the cases, and studying affected children can provide valuable information about the inheritance patterns, genetic mutations, and molecular mechanisms underlying the condition. This knowledge can have implications not only for pediatric patients but also for their families, enabling genetic counseling and potentially identifying individuals at risk within the same family.

The early identification of at-risk individuals, such as family members of affected children, may allow for targeted screening and interventions to prevent or delay the onset of ARVC-related complications. By focusing on this specific population, researchers and healthcare professionals can make significant strides in improving the diagnosis, management, and outcomes of children affected by ARVC.

Therefore, we present a literature review on arrhythmogenic right ventricular dysplasia, aiming to identify the main features of the disease in the pediatric population: diagnosis age, presentation patterns, electrocardiogram (ECG), echocardiography and magnetic resonance imaging (MRI) findings, as well as genetic analysis, heredocolateral antecedents and outcomes of the disease.

## 2. Methods

We performed an electronic search in PubMed and Embase databases from the time of their inception to October 2022 using the following keywords: [“arrhythmogenic right ventricular dysplasia” OR “arrhythmogenic right ventricular cardiomyopathy” OR “ARVD/C”] AND [“children” OR “pediatric” OR “paediatric”]. The research retrieved 392 results in PubMed and 497 in Embase. After removing the duplicates, 279 studies were screened for eligibility by title and abstract. We aimed to include all the reports that include details about the following characteristics: diagnosis age, reasons of presentation, electrocardiography patterns and, if possible, echocardiographic and magnetic resonance imaging findings, genetic test results, and follow-up patterns.

We excluded 239 papers due to the following reasons: 118 studies exclusively on adults, 26 clinical protocols, 24 conference abstracts, 17 articles on other cardiac conditions (tricuspid atresia, heart transplant), 17 in vitro studies, 14 letters to editor, 12 clinical trials on various treatments, 8 studies on animals, and 3 consensus papers. From the remaining 40 papers, we further excluded 31 papers, as 18 were limited to only genetic testing or treatment efficiency, and 13 consisted of single case-reports. Finally, 9 papers were included for analysis (Figure 1).

## 3. Results

Several characteristics of the included studies were detailed: sample size, male-to-female ratio, age at diagnosis, reasons for presentation, electrocardiography, echocardiographic and magnetic resonance imaging findings, genetic test results, and follow-up patterns (Table 1).

### 3.1. Clinical Characteristics and Follow-Up

Our analysis included 593 patients aged 3 to 17.4 years. Of those, only 50 were in the asymptomatic phase of the disease. The asymptomatic cases were incidentally discovered via abnormal ECG or echocardiographic findings.

In symptomatic cases, the severity of symptoms ranged from palpitations to heart failure and cardiac arrest. In the study conducted by Chungsomprasong et al. [8] and Etoom et al. [11], most of the cases were admitted with cardiac symptoms. However, the authors did not specifically define the patterns of these clinical characteristics. On the other hand, three of the included studies [9,15,16] clearly detailed the manifestations at presentation. Deshpande et al. [9] reported dizziness and syncope in 37.5% of the children diagnosed with ARVC, Surget et al. [15] noted that 40% of the children presented with palpitation, whereas Te Riele et al. [16] described that 37.3% of the included cases were characterized by presyncope, syncope, and/or palpitations.

In Surget et al.’s study [15], the age at diagnosis influenced the evolution of the cases: children diagnosed before puberty presented more frequently with a biventricular or left-dominant arrhythmogenic cardiomyopathy (LD ACM) and had a poorer prognosis due to severe heart failure (48% vs. 10%). No relationship with age was reported in all the other papers.

Also, 6/13 of the included studied assessed the outcome of the patients, showing that the outcome of patients ranged from episodes of nonsustained ventricular tachycardia to heart failure, heart transplant, and sudden cardiac death. Among these, sudden cardiac death was responsible for the most severe scenarios in various proportions ranging from 0/12 [13] to 8/16 [9]. On the other hand, Chungsomprasong et al. [8] reported no significant difference in the rate of deterioration of biventricular ejection fraction or of chamber enlargement revised Task Force Criteria categories.

### 3.2. ECG and Imaging Studies

The diagnosis of ARVC is challenging, as no single imaging study is sufficiently sensitive or specific to establish a definitive diagnosis. All patients had electrocardiography abnormalities ranging from T wave inversion in chest leads (persisting after 11 years of age and/or extending beyond V3), terminal activation delay in right chest leads, atypical RBBB, and ventricular ectopy with LBBB morphology (especially if from RV free wall) to ventricular tachycardia and ventricular fibrillation. Echocardiography findings included decreased circumferential ventricular strain, right ventricular dilatation with regional wall motion abnormalities and systolic dysfunction, dyskinetic bulges, right ventricular output tract dilatation, and right ventricular wall thinning. The importance of magnetic resonance imaging for diagnosis was best described by Etoom et al. [11] who underlined that this diagnostic tool is more useful than echocardiography in the context of revised Task Force Criteria.

From an electrocardiography point of view, the 2023 ESC Guideline on cardiomyopathies shows that late potentials have poor specificity and sensitivity and, therefore, add little value to the diagnostic inquiry, as do epsilon waves that are more frequently found if the right ventricle shows significant structural abnormalities. On the other hand, every suspected ARVC patient should undergo yearly Holter monitoring, as this investigation has shown risk stratification utility.

Although not widely available worldwide, positron emission tomography using 18-fluorodeoxyglucose can show myocardial uptake, but careful differential diagnosis should be made with cardiac sarcoidosis, as well as testing positive with this type of investigation [17].

### 3.3. Genetic Analysis and Histopathological Findings

Family background was positive in most patients in the included studies, whereas genetic testing was performed in 7/9 cases showing various mutations in genes encoding cardiac desmosome proteins (e.g., *DSG2*, *PKP2*, *DSP*).

Histopathological findings were detailed in 3/9 of the included studies [9,18,19]. Deshpande et al. [9] demonstrated the fibro-fatty replacement of right ventricular myocardium in all cases by using endomyocardial biopsies. In Etoom et al.’s study [11], only 26% of the subjects underwent endomyocardial biopsy, with a positive biopsy in 6 of 9 patients with definite ARVC and 3 of 11 with borderline ARVC. Fogel et al. [12] also reported fibrofatty infiltration in four of nine patients who met the Task Force Criteria.

## 4. Discussions

Our systematic review provides an overview of the current evidence regarding ARVC in children. The review’s findings may have implications for clinical practice by contributing to the understanding of the disease, guiding diagnostics, identifying research gaps, and supporting patient education and counseling. Understanding the natural history of the disease and the factors that influence disease progression can aid in risk stratification and personalized management plans for affected children. The positive and differential diagnosis, as well as risk stratification and future directions of research, may be useful for healthcare providers.

### 4.1. From Clinical Suspicion to Genetic Confirmation

According to Elias Neto et al. [20], the symptoms of ARVC/D usually occur between the third and fifth decades of life as ventricular arrhythmia episodes that may progress to sudden cardiac death. In children, case reports revealed that the most common sign was ventricular tachycardia which frequently presented as dizziness, syncope, or even cardiac arrest in children aged 3 to 16 years [9]. However, it is not uncommon for children with ARVC/D to be asymptomatic, although healthy carriers can still transmit the disease to their descendants [21].

The natural history of ARVC/D was classified by Pinamonti et al. [22], according to the progression of structural alterations and clinical manifestations, as follows: 1. occult phase (subclinical phase)—asymptomatic—but with possible discrete structural abnormalities in the right ventricle; associates risk of sudden death; 2. arrhythmic phase—palpitations and/or syncope triggered by physical effort due to various types of arrhythmias ranging from isolated ventricular ectopic beats (non-sustained ventricular tachycardia) with left bundle branch block morphology to ventricular fibrillation, which may be fatal; 3. right ventricular failure due to the replacement of myocardial tissue with fibro-fatty tissue; 4. biventricular failure—may associate right ventricle aneurysms in the presence of atrial fibrillation, which leads to mural thrombosis. An early diagnosis is essential for a better prognosis because the risk of sudden death is cited even in the occult phase. Recently, researchers in Sweden reported that 27% (6/22) of the cases of autopsy-confirmed ARVC/D experienced no symptoms in the six months preceding sudden cardiac death [23]. The evolution of the disease depends on cardiac electrical instability and the progressive deterioration of cardiac performance. Adrenergic tone may contribute to arrhythmogenesis because it increases the susceptibility to ventricular arrhythmias, especially during high-intensity exercise [24]. The progressive deterioration of ventricular performance may lead to right or biventricular heart failure.

The first diagnostic criteria for ARVC/D date back to 1994 when “the Task Force of the Working Group of Myocardial and Pericardial Diseases of the European Society of Cardiology and Scientific Council on Cardiomyopathies of the International Society and Federation of Cardiology” proposed a list of major and minor criteria including imagistic and electrocardiographic patterns [25]. As these criteria lacked a precise quantitative stratification regarding the grade of fibrosis or right ventricle dilatation/dysfunction, with arrhythmic features included in the minor criteria category, in 2010, the Task Force proposed an updated guideline providing quantitative criteria for the diagnosis of right ventricle abnormalities and also molecular genetic criteria [26].

Deshpande et al. [9] underlined the importance of adapted criteria for pediatric cases, as the adherence to the Task Force guideline, although recommendable, may lead to misdiagnosis in children with early fibro-fatty replacement of the right ventricular myocardium. In a study including 48 children with suspected ARVC/D, Steinmetz et al. [14] reported that children with positive genetic findings could be identified reliably by combining echocardiography and contrast-enhanced magnetic resonance imaging. The use of cardiac magnetic resonance imaging has proven effective in identifying certain cases that were not detectable through echocardiography [27]. As a result, magnetic resonance plays a crucial role in correct diagnosis in individuals with suspected ARVC/D.

A 2019 International Expert report evaluated these previous criteria, concluding that although characterized by good accuracy, these criteria did not prove to have enough sensitivity for the identification of the expanding phenotypic disease spectrum, which also includes left-sided variants [28]. In 2020, the “Padua criteria” were proposed for both right- and left-sided arrhythmogenic cardiomyopathy phenotypes, and recently [29], Corrado et al. [30] refined these criteria, incorporating the myocardial scar criteria by using the late-gadolinium enhancement technique for a complete characterization of right, biventricular, and left disease variants. Furthermore, the 2023 ESC guideline on cardiomiopathies favors magnetic resonance imaging over other imaging methods to make the diagnosis [17].

A histological analysis of the cardiac tissue from children with ARVC identified a number of important characteristics. One of the key findings was that fibrofatty tissue, which is defined by the presence of adipocytes and fibrous connective tissues, replaces the normal myocardium. In addition to fibrofatty replacement, other histological findings include myocyte loss, myocyte hypertrophy, and fibrosis. Fibrotic areas, which can be observed within the fibrofatty tissue, may contribute to the disruption of normal myocardial architecture and serve as arrhythmogenic substrates, facilitating the occurrence of ventricular arrhythmias, a hallmark of ARVC. The histopathological findings in pediatric ARVC may differ from those observed in adult cases. While fibrofatty replacement is a common feature in both populations, pediatric cases tend to exhibit a higher prevalence of myocyte hypertrophy and fibrosis [31]. These differences may reflect the unique pathophysiological processes underlying ARVC development in children.

However, currently, there is no gold standard for diagnosing ARVC/D in children. Since a definite diagnosis is usually based on significant alterations not apparent in the early phases of the disease, genetic analysis may be a proper diagnostic tool for asymptomatic cases. The majority of the ARVC/D pathogenic mutations affect genes encoding structural proteins involved in the organization of intercellular junctions. Corrado et al. [28] reported that genetic alteration in at least one of the genes (*DSC2*, *DSG2*, *PKP*, *JUP*, *DSP*) encoding cardiac desmosome proteins is presented in nearly 60% of ARVC/D patients. Most ARVC/D cases are autosomal-dominant with incomplete penetrance [32]. The autosomal recessive pattern of inheritance has also been reported as Naxos disease (ARVC, palmoplantar keratoderma, and woolly hair) [33], with higher prevalence in higher in Italy (Padua, Venice) and Greece (Island of Naxos) [34].

Genotyping is not only beneficial for diagnostic purposes, but also for prognostic reasons, as some genes are associated with a higher risk of sudden cardiac death and heart failure. While the role of desmosomal genes in ARVC has been extensively studied, there is growing evidence suggesting the involvement of non-desmosomal genes. According to Bosman et al. [35], ARVC may be caused by mutations in desmosome, as well as adherens junctions, cytoskeleton, cytokines, and ion transporters. Mutations of CTNNA3 and CDH2 affect the adherens junction but are rarely associated with ARVC [35]. The cytoskeleton can also be affected. For instance, mutations of *TMEM43 p.S358L* were associated with a higher disease penetrance and risk of sudden cardiac death [36]. DES, which encodes the protein desmin, plays a crucial role in maintaining the structural integrity of cardiac muscle cells [37]. Mutations in DES have been identified in a subset of ARVC patients, indicating its potential contribution to the pathogenesis of the disease [38]. These mutations can disrupt the normal organization of the cytoskeleton and impair cellular function, ultimately leading to the development of ARVC [39]. Van Tintelen et al. investigated the clinical characteristics of 27 patients from five families with an identical mutation in the head domain region (p.S13F) of desmin, reporting that all these patients developed high-grade atrioventricular block at young ages and important right ventricular involvement [40]. Mutations in ILK, an integrin-linked kinase, which is involved in cell adhesion, signaling, and cytoskeletal organization, can also contribute to the development of ARVC [41]. LEMD2, a gene involved in nuclear envelope organization and chromatin regulation, has also emerged as a potential mutation involved in ARVC [18]. Mutations in PLN were associated with altered intracellular calcium flow with increased susceptibility to arrhythmias [19]. Consequently, the use of genetics in the correct stratification of arrhythmic risk is increasingly being promoted, and recent discoveries support the importance of molecular analysis in the diagnosis of these patients. In ARVD/C cases with a negative desmosomal genetic analysis, variants in the gene encoding the giant sarcomeric protein called titin have also been reported [42,43]. However, more recently, the validity of the association between titin variants and ARVC/D has been under debate [3,44]. Martínez-Barrios et al. [45] reanalyzed the role of titin variants in inherited arrhythmogenic syndromes and concluded that most missense titin variants had no deleterious role in ARVD/C, admitting that additional studies are required to clarify this association.

### 4.2. Differential Diagnosis: Mimicking Conditions and Phenotypes

Several arrhythmic diseases and structural conditions involving the ventricular myocardium may represent a diagnostic trap. Idiopathic right ventricular outflow tract tachycardia, athlete’s heart, Brugada’s syndrome, dilated cardiomyopathy, cardiac sarcoidosis, myocarditis, and congenital ventricular outpouchings (aneurysm/diverticula) are the main differential diagnosis for ARVC/D [46]. Distinguishing ARVD/C from idiopathic right ventricular outflow tract dilatation may be challenging, but the latter is a benign condition characterized by the absence of structural abnormalities [47].

Cardiac sarcoidosis mimicking definite ARVC/D has also been described [19,48,49]. The diagnosis of cardiac sarcoidosis is supported by prolonged PR interval, advanced atrioventricular block, longer QRS duration, right ventricular apical involvement, low left ventricular ejection fraction, and positive 18F-FDG PET scan, whereas larger right ventricular outflow tract dimensions, subtricuspid involvement, and peripheral T-wave inversion are in favor of ARVC/D [42]. Also, sarcoidosis is associated with an older age of symptom onset, non-familial disease pattern, and mediastinal adenopathy [50].

Differential diagnosis between ARVC/D and myocarditis may be challenging for clinicians as bouts of acute or subacute myocarditis may sometimes occur in ARVC/D [51]. In these cases, advanced cardiovascular magnetic resonance and endomyocardial biopsy may guide the clinician to the correct diagnosis [52]. ARVC/D may also mimic dilated cardiomyopathy, especially in cases with predominantly left ventricle involvement, but ARVD/C is characterized by significant ventricular arrhythmias disproportionate to the left ventricular systolic dysfunction [53].

Recently, Bariani et al. [54] described an acute phase of arrhythmogenic cardiomyopathy characterized by chest pain, release of biochemical markers of myocardial necrosis, and electrocardiographic changes in the absence of coronary artery disease. The pathogenetic mechanism of this so called “hot phase” of the disease is unclear, but it seems that the inflammatory process may be a trigger for subsequent tissue necrosis and replacement with fibroadipose tissue. In a clinical setting that mimics myocarditis, a family history of arrhythmogenic cardiomyopathy and positive genetics should support the differential diagnosis.

### 4.3. Risk Stratification and Prognosis

Risk stratification in patients with ARVD/C may be beneficial, even in asymptomatic individuals, because sudden cardiac death can be the first presentation of ARVD/C. A recent analysis by The European Society of Cardiology established that the most relevant predictors of life-threatening arrhythmic events, including sudden cardiac death, include the following: right ventricular dysfunction and syncope, younger age, QRS fragmentation, and non-sustained ventricular tachycardia [55]. However, there is currently no available risk prediction model for risk stratification adapted for children.

The value of genetic testing in the diagnosis and risk stratification of arrhythmogenic right ventricular cardiomyopathy was underlined by de Brouwer et al. [56] in a multicenter cohort study including 402 ARVD/C cases: the diagnosis of ARVD/C would have been missed or delayed in 10% of patients with ARVC (40 of 402) if the genetic criterion of the Task Force Criteria had been disregarded, but malignant variants of ventricular arrhythmias occurred in only 1% of cases with lost or delayed diagnoses (3 of 402). According to Wallace et al. [57], the estimation of predictive risk factors of ventricular and sudden cardiac deaths is based on these factors: age of onset, male sex, specific genetic mutations, cardiac syncope, history of ventricular arrhythmias, degree of myocardial involvement, electrical instability, and exercise restriction.

Recently, Surget et al. reported a high prevalence of heart failure in children aged less than 12 years as a first clinical manifestation (37% of cases), in comparison with post-pubertal-age children whose initial presentation consisted of ventricular tachycardias [15]. Roudijk et al. [13], in a study including 12 probands with pediatric-onset ARVC/D and 68 pediatric relatives (aged <18 years at first evaluation), found that sudden cardiac death may be the first manifestation of ARVD/C in both pediatric probands and relatives, supporting the importance of imagistic and electrocardiographic monitoring.

In arrhythmogenic cardiomyopathy, the mechanism of sudden cardiac death is represented in approx. 90% by sustained monomorphic ventricular tachycardia (although expressing pleomorphism due to multiple circuits) and in approx. 10% of cases by ventricular fibrillation/polymorphic ventricular tachycardia. Malignant ventricular arrhythmias may occur even as a first presenting sign of the disease [3]. In a comprehensive meta-analysis conducted by Bosman et al., the most significant predictors of ventricular arrhythmias in patients with definite ARVC were represented by the following factors: male sex, unexplained syncope, T-wave inversion in the V3 lead, right ventricular dysfunction, and previously registered (non)sustained ventricular tachycardia or ventricular fibrillation [58]. In cases with borderline ARVC, two supplemental predictive factors were identified: inducibility during electrophysiological study and strenuous exercise [58]. In mutation carriers, in addition to the predictors mentioned above, ventricular ectopy, multiple ARVC-related pathogenic mutations, left ventricular dysfunction and palpitations/presyncope were the main determinants for the arrhythmic risk [58]. In 2019, a score for sudden cardiac death risk stratification was proposed (the 2019 ARVC risk score, www.arvcrisk.com) according to the conclusions of the largest cohort of ARVC patients with no sustained ventricular arrythmias history at diagnosis [59]. The variables included were as follows: sex, age, cardiac syncope (defined as the transient loss of consciousness and postural tone with spontaneous recovery with a likely arrhythmic origin, which occurred in the prior 6 months), non-sustained ventricular tachycardia (NSVT), the number of premature ventricular complexes on 24 h Holter monitoring, the extent of T-wave inversion on anterior and inferior leads, and the proper ventricular ejection fraction [59]. Despite several drawbacks, the suggested model can accurately predict the prognosis of ARVC patients who do not have a history of sustained ventricular arrhythmias at the moment of diagnosis.

### 4.4. Future Directions of Research

Further investigation into the genetic basis of ARVD in children can help identify specific gene mutations or variations that contribute to the development of the condition. This may contribute to early diagnosis, risk assessment, and personalized treatment approaches. Developing reliable biomarkers and diagnostic tools specific to ARVD in children can improve early detection and accurate diagnosis. This may involve exploring novel imaging techniques, genetic testing methods, or blood-based biomarkers.

Research efforts may focus on refining risk stratification models to better predict the progression and outcomes of ARVD in children. This can help guide treatment decisions and identify individuals who may benefit from early interventions or closer monitoring.

Another important area of research is the innovative treatment options and interventions. This may involve studying the effectiveness of medications, surgical approaches, or novel therapies such as gene therapy or stem cell transplantation.

The long-term outcomes and impact of ARVD on the quality of life in children are also important. Understanding the factors that influence disease progression, cardiac function, and psychosocial well-being may optimize management strategies and support the patients and their families.

## 5. Conclusions

Despite the fact that arrhythmogenic right ventricular dysplasia can remain asymptomatic for a variable period of time, this condition can often prove fatal. In symptomatic cases, the severity of symptoms ranges from palpitations to heart failure and cardiac arrest. The diagnosis of ARVC is challenging, as no single imaging study is sufficiently sensitive or specific to establish a definitive diagnosis. When the disease becomes symptomatic (arrythmia, right ventricular or biventricular failure), an experienced cardiologist may raise the suspicion of this diagnosis based on echocardiography. In selected cases, the indication of performing cardiac magnetic resonance imaging can bring additional details which echocardiography may not be able to detect.

The next step is genetic testing, and once the disease is confirmed, genetic counseling should be offered to the family. Genotyping is not only beneficial for diagnostic purposes, but also for prognostic reasons, as some genes are associated with a higher risk of sudden cardiac death and heart failure.

The outcome of patients ranges from episodes of nonsustained ventricular tachycardia to heart failure, heart transplant, and sudden cardiac death. In selected cases, the patient can benefit from appropriate treatment to prevent sudden cardiac death.

## Figures and Tables

**Figure 1 diagnostics-14-00175-f001:**
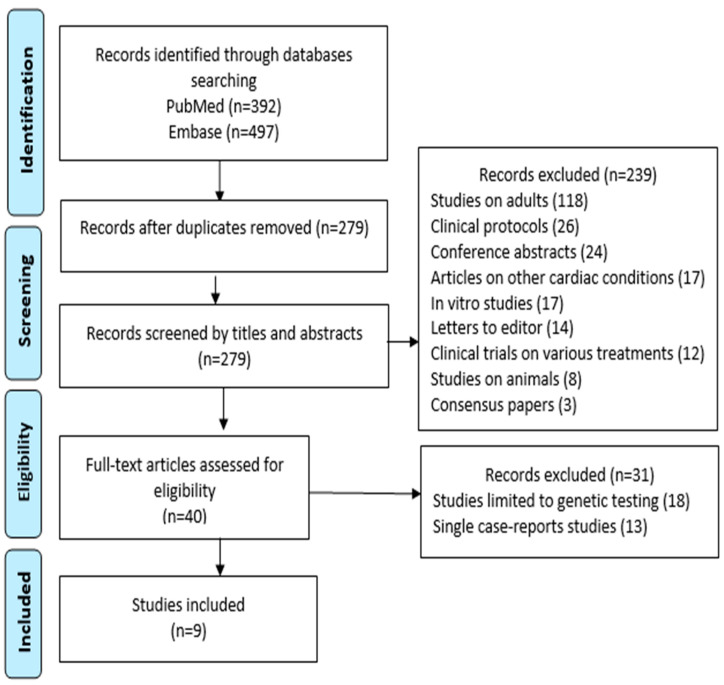
Prisma flow-chart.

**Table 1 diagnostics-14-00175-t001:** Main studies reporting ARVC/D cases in pediatric population.

Study (Author, Year)	Study Sample (Male:Female Ratio)	Age at Diagnosis	Presentation	ECG Findings	Echocardiography Findings	MRI (Magnetic Resonance Imaging) Findings	Genetic Testing	Family Background	Outcome
**Chungsomprasong et al., 2017 [8]**	142 cases (80:62)	3 to 18 years (median 14.6 years)	55/142—cardiac symptoms, 49/142—ventricular arrhythmias, 3/142—incidentally discovered abnormal ECG findings, 5/142—incidental abnormal echocardiographic findings	49/142— ventricular arrhythmias	Not detailed (MRI—based study)	Higher rTFC scores correlated with lower right ventricular EF and lower left ventricular global circumferential strain	8/39: 5 *PKP-2*, 1 *desmocollin-2*, 1 *DSP*, 1 transmembrane protein-43)	Positive in 67/142 cases	The rate of deterioration of biventricular EF or of chamber enlargement did not differ between rTFC categories.
**Deshpande et al., 2016 [9]**	16 cases (10:6)	3 to 16 years (mean 12.6 years)	5/16—cardiac arrest, 6/16—VT (dizziness, syncope) 2/16—heart failure	11/16 patients: ventricular ectopy, first-degree heart block, LBBB, non-sustained VT, ventricular fibrillation	Right ventricular dilatation and dysfunction in all ECG suggestive cases	RV dilatation, regional wall motion abnormalities, and systolic dysfunction (in 3/4 investigated cases)	3/16: *DSG2*, *TMEM43*, and *RYR2* mutation	Positive in 5/16 cases	8/16—SCD, 7/16—alive (4 with HT), 1/16—lost to follow-up
**DeWitt et al., 2019 [10]**	16 cases (10:6)	14.9 ± 2.0 years	3/16—cardiac arrest, 7/16—VT, 2/16—arrhythmic syncope, 1/16 –palpitations, 4/16—asymptomatic	Not detailed, but ECG features were either present before or concurrently with structural alterations that were visible in cardiac imaging results. This suggests that the ability of cardiac imaging to detect early disease manifestation is limited when ECG features are absent.			14/16: *PKP2*	Positive in 14/16 cases	2/16 -nonsustained VT during follow-up
**Etoom et al., 2015 [11]**	142 cases (80:62)	13.8 ± 3.2 years	81/142—cardiac symptoms, 75/142— ventricular arrhythmias, 4/142—incidentally discovered abnormal ECG findings, 6/142—incidental abnormal echocardiographic findings, 100/142—family history	54/142—depolarization abnormalities, 9/142—repolarization abnormalities, 26/142—arrhythmias	44/94: right ventricle wall motion abnormalities	11/23 with definite ARVC would not have been in this group if CMR had not been performed; CMR is more useful as a diagnostic imaging tool than echocardiography for ARVC diagnosis in the context of the rTFC	8/44 positive genetic testing (mutations not mentioned)	Positive in 100/142 cases	Not assessed
**Fogel et al.,** **2006 [12]**	81 cases (46:35)	11.5 ± 5.5 years	16/81—VT or fibrillation, 34/81—tachycardia, palpitations, or PVC, 15/81—cardiac arrest, 16/81—asymptomatic	52/81—abnormal findings: 29 prolonged QRS, 3 ɛ waves, 4 right atrial enlargements, 9 right ventricle hypertrophies, 7 LBBBs, 3 RBBBs	3/81 referred due to dilated and poorly functioning right ventricle	2/81— fatty infiltrations of the right ventricle, 2—right atrial and right ventricle dilation, 3/81—right ventricle dyskinesia or dyskinetic bulges, 2/81—RVOT ectasia, 2/81—RV wall thinning	Not performed	Positive in 26/81 cases	Not assessed
**Roudijk et al., 2021 [13]**	12 cases (8:4)	13.8–17.4 years (mean 16.6 years)	7/12—VT, 1/12—SCD, 2/12—resuscitated cardiac arrest, 2/12—(near)-syncope or palpitations	9/12—T-wave inversion V1-V3, 1/12—T-wave inversion V1-V2, 2/12—prolonged terminal activation duration	5/12—abnormal cardiac imaging, with a median RVEF LVEF of 33% (25%–45%) and 49% (39%–56%), respectively	3/4—late gadolinium enhancement	9/11: 5 *PKP2*, 4 multiple pathogenic variants	12/68 diagnosis at <18 years old, 44/47—(likely) pathogenic variant	19 (1 proband and 18 relatives) diagnosed with ARVC during follow-up; 0 SCDs
**Steinmetz et al., 2018 [14]**	48 cases (32:16)	12.9–15.1 years (mean 14 years)	Ventricular ectopic beats, VT and positive family history (ARVC and/or sudden cardiac death within the family (number not specified))	Not detailed	Not detailed	Not detailed	12/48: 9 *DSP*, 3 *PKP 2*	Not detailed	Not assessed
**Surget et al., 2022 [15]**	61 cases (51:10)	21 cases in group 1 (8 ± 3 years), 40 cases in group 2 (15 ± 2 years)	40%—palpitations, 26%—syncope, 3%—resuscitated cardiac arrest; children diagnosed before puberty presented more frequently with a biventricular or LD ACM and had a poorer prognosis due to severe HF (48% vs. 10%)	Group 2 presented more VT as initial presentation (61% vs. 24%, *p* = 0.007), but during the follow-up, there was no difference in VT occurrence between the 2 groups (45% versus 57%, *p* = 0.4).	Not detailed	Not detailed	26/37: 6 *DSP*, 14 *PKP2*, 3 *DSG2*, 1 *desmin*, 1 *SCN10A*, 1 *lamin A/C*	Positive in 15/61 cases	33% had transplantation or died secondary to HF in group 1 versus 3% in group 2
**Te Riele et al., 2015 [16]**	75 cases (41:34)	15.3 ± 2.4 years (median 15.9 years)	19/75—SCD, 16/75—sustained VT, 28/75—(pre-) syncope and/or palpitations, 12—asymptomatic	31/45—sustained monomorphic VT (83% LBBB, 11% RBBB, and 6% polymorphic VT)	52/75 abnormal imaging results (not detailed)	52/75 abnormal imaging results (not detailed)	Not performed	6/20—SCD	11/75—SCD

ECG—electrocardiography, HF—heart failure, HT—heart transplantation, LBBB—left bundle branch block, LD ACM—left-dominant arrhythmogenic cardiomyopathy, PVC—premature ventricular contractions, RBBB—right bundle branch block, rTFC—revised Task Force Criteria, EF—ejection fraction, CMR—cardiac magnetic resonance, RVEF—right ventricular ejection fraction, SCD—sudden cardiac death, VT—ventricular tachycardia.

## Data Availability

Data is contained within the manuscript.

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
