# Peer review of "Arrhythmogenic Right Ventricular Cardiomyopathy in Children: A Systematic Review"

_diagnostics, 2024, doi:10.3390/diagnostics14020175_

Round 1

Reviewer 1 Report

Comments and Suggestions for Authors

In the review articel 'Arrhythmogenic right ventricular cardiomyopathy in children' submitted by Stefana Maria Moisa et al. the authors discussed the clinical impact of ARVC for children.

The topic of this review article is interesting. However, the manuscript needs severeal extensions and some minor changes.

1.) Line 32: ARVC instead ARV

2.) Introduction: The authors are explaining that ARVC is mainly caused by fibro-fatty replacement of the myocardium. However, I would also discuss that the left ventricle is also affected at a later stage of the disease. For example, the fibro-fatty replacement within the left-ventricular myocardial tissue received from a heart transplantation was analyzed recently by spatial transcriptomics. 

3.) The inclusion and exclusion criteria need revision and further explanations. For example, a study about hemi and homozygous DSG2 mutations is not mentioned Table 1 , although the authors have described an ARVC patient receiving its diagnosis at the age of 12. Since genetic, histology, molecular and clinical data are described, I would add this to this review article (10.3390/ijms22073786) . 

4.) Gene names should be written in Italics. 

5.) The authors discussed mutations in the non-desmosomal gene TTN, encoding TTN (Line 261 following). I think the authors should also discuss other relevant non-desmosomal genes like DES (desmin), ILK (integrin linked kinase) and LEMD2. Especially for DES, there are several reports justifying that this gene should be included in a systematic review article about ACM. This is also relevant, since the desmin filaments are connected to the desmosomes, which is the major cellular structure affected in ACM (ARVC). 

6.) ‘genetic etiology’ instead of ‘genetic component’ (line 70)

7.) gene not genes (singular, line 49)

8.) I suggest to add "early onset" to the filter keywords. I think you missed several important reports.

Nevertheless, I think that the topic of this review article is really interesting but needs several extensions and some minor changes. Good luck with the revision.

Author Response

Dear Sir/Madam,

We respectfully thank you for your review. We hereby provide a point-by-point response to all the issues pointed. The changes that we did have been included in the main manuscript with tracked changes in order to improve legibility.

1.) Line 32: ARVC instead ARV

We corrected, thank you for the observation.

2.) Introduction: The authors are explaining that ARVC is mainly caused by fibro-fatty replacement of the myocardium. However, I would also discuss that the left ventricle is also affected at a later stage of the disease. For example, the fibro-fatty replacement within the left-ventricular myocardial tissue received from a heart transplantation was analyzed recently by spatial transcriptomics.

We acknowledge that ARVC is a diffuse disease involving both ventricles (and atria also!), and especially in advanced phases LV and atria involvement become apparent. We modified accordingly the introduction

3.) The inclusion and exclusion criteria need revision and further explanations. For example, a study about hemi and homozygous DSG2 mutations is not mentioned Table 1 , although the authors have described an ARVC patient receiving its diagnosis at the age of 12. Since genetic, histology, molecular and clinical data are described, I would add this to this review article (10.3390/ijms22073786).

We respectfully remind to the reviewer that it is impractical to perform a systematic review analyzing all the data scattered from single case reports. We cannot modify the inclusion and exclusion criteria just to accommodate case reports such as those suggested by the (hidden) reviewer.

4.) Gene names should be written in Italics.

We corrected, thank you for the observation.

5.) The authors discussed mutations in the non-desmosomal gene TTN, encoding TTN (Line 261 following). I think the authors should also discuss other relevant non-desmosomal genes like DES (desmin), ILK (integrin linked kinase) and LEMD2. Especially for DES, there are several reports justifying that this gene should be included in a systematic review article about ACM. This is also relevant, since the desmin filaments are connected to the desmosomes, which is the major cellular structure affected in ACM (ARVC).

We updated the description of non-desmosomal gene mutations, thank you for the observation.

6.) ‘genetic etiology’ instead of ‘genetic component’ (line 70)

We modified to ,,in more than 60% of cases a pathogenic/likely pathogenic gene variant is described”, thank you for the observation.

7.) gene not genes (singular, line 49)

We corrected, thank you for the observation.

8.) I suggest to add "early onset" to the filter keywords. I think you missed several important reports.

We respectfully mention to the reviewer that children is quite ,,early”, especially for a disease that usually is diagnosed in young adults.

Reviewer 2 Report

Comments and Suggestions for Authors

The paper “ Arrhythmogenic right ventricular cardiomyopathy in children. A systematic reviewis interesting presenting the data on this rare and challenging anomaly, based upon a review of the current literature.

The authors found eligible from the PubMed and Embase databases (as evident on the Prisma fig.) 9 papers that presented outcomes and summarized these data in a comprehensive Table giving numbers of patients in each study, relative data (age, presentation, etc. ) and relative outcomes); on the whole 593 patients.

I find the paper very interesting and I have only small points to clear:

Page 2 – line 62 …treatement is crucial

Line 80..We present (better than proporse)

Page 3 – line 111 Clinical characteristics: the references 11,12,13 and 14 ar4 not commented in the description – to be completed.

Page 3 line  124 – the abbreviation LDACM should be at first described as a whole name (as then evident in the Legend of the Table.

Page 3, lin 136 . “Moreover”- it would be better to say “On the other hand

Paragraph 3.3 Genetic analysis – lines 164 …– the references 18-39 are not mentioned

Line 153 – cardiomyopathy .

Line 185 and further in the discussion  – It would be better to put the reference number of the authors immediately after the name..

I am suggesting to do the small corrections – as minor revision.

Author Response

Dear Sir/Madam,

We respectfully thank you for taking the time to review our manuscript and for your suggestions. We modified the manuscript accordingly.

We provide a point by point response to your suggestions.

Page 2 – line 62 …treatement is crucial

We believe the structure of the phrase could remain unchanged- “making accurate diagnosis and effective treatment crucial for their well-being”

Line 80..We present (better than proporse)

Done! Thank you!

Page 3 – line 111 Clinical characteristics: the references 11,12,13 and 14 ar4 not commented in the description – to be completed.

References 8-16 are listed in order of their appearance in the table. Furthermore,

Reference 11 is quoted at line 170

Reference 12 is quoted at line 116

Reference 13 is quoted at line 136

Reference 14 is quoted at line 225

Page 3 line  124 – the abbreviation LDACM should be at first described as a whole name (as then evident in the Legend of the Table.

Done! Thank you!

Page 3, lin 136 . “Moreover”- it would be better to say “On the other hand

Done!

Thank you!

Paragraph 3.3 Genetic analysis – lines 164 …– the references 18-39 are not mentioned

Reference 18 is mentioned at line 188

Reference 19 is mentioned at line 192

And so on…

Reference 39 is mentined at line 281

Line 153 – cardiomyopathy .

Done! Thank you!

Line 185 and further in the discussion  – It would be better to put the reference number of the authors immediately after the name..

Done!

Thank you!

Round 2

Reviewer 1 Report

Comments and Suggestions for Authors

The authors have addressed my points in a sufficient way. I suggest to accept this manuscript for publication.